# MCUFormer: Deploying Vision Transformers on Microcontrollers with Limited Memory

Yinan Liang[1]    Ziwei Wang[2]    Xiuwei Xu[2]    Yansong Tang[1,*]    Jie Zhou[2]    Jiwen Lu[2]

{[1]Shenzhen International Graduate School, [2]Department of Automation}, Tsinghua University

## Abstract

Due to the high price and heavy energy consumption of GPUs, deploying deep models on IoT devices such as microcontrollers makes significant contributions for ecological AI. Conventional methods successfully enable convolutional neural network inference of high resolution images on microcontrollers, while the framework for vision transformers that achieve the state-of-the-art performance in many vision applications still remains unexplored. In this paper, we propose a hardware-algorithm co-optimizations method called MCUFormer to deploy vision transformers on microcontrollers with extremely limited memory, where we jointly design transformer architecture and construct the inference operator library to fit the memory resource constraint. More specifically, we generalize the one-shot network architecture search (NAS) to discover the optimal architecture with highest task performance given the memory budget from the microcontrollers, where we enlarge the existing search space of vision transformers by considering the low-rank decomposition dimensions and patch resolution for memory reduction. For the construction of the inference operator library of vision transformers, we schedule the memory buffer during inference through operator integration, patch embedding decomposition, and token overwriting, allowing the memory buffer to be fully utilized to adapt to the forward pass of the vision transformer. Experimental results demonstrate that our MCUFormer achieves 73.62% top-1 accuracy on ImageNet for image classification with 320KB memory on STM32F746 microcontroller. Code is available at https://github.com/liangyn22/MCUFormer.

## 1 Introduction

Deep neural network deployment for realistic applications usually requires high-performance computing devices such as GPUs and TPUs [16, 12]. Due to the high price and energy consumption of these devices, the unacceptable deployment expenses strictly restrict the utilization of deep models in a widely variety of tasks[17, 39], especially in the scenarios without sufficient battery support. Deploying deep neural networks on cheap IoT devices with low energy consumption becomes a practical solution in many industrial applications such as maritime detection [34, 24], agricultural picking [31, 43] and robot navigation [2, 29].

Recently, the hardware-algorithm co-design framework [20, 19, 30] is widely studied to deploy deep convolutional networks with high storage and computational complexity to microcontroller units (MCU) with extremely limited memory (SRAM) and storage (flash). By searching the optimal architectures and arranging the memory buffer according to the overall network topology, deep convolutional neural networks with large numbers of parameters and high FLOPs are successfully deployed on MCUs with comparable accuracy as inferencing on high-performance computing devices. Vision transformers [22, 33] have surpassed convolutional neural networks in a wide variety of tasks, which can further push the limit of intelligent visual perception in resource-constrained devices.

---

*Corresponding author.

37th Conference on Neural Information Processing Systems (NeurIPS 2023).

However, the memory footprint of vision transformers significantly exceeds the budget of MCUs due to the matrix multiplication with large sizes, and the lack of operator library for transformer inference on MCUs also obscure the practical deployment.

In this paper, we present a hardware-algorithm co-optimizations framework to deploy vision transformers on MCUs with extremely limited memory budget. We jointly design transformer architectures that achieve the highest task performance within the memory resource constraint and construct an operator library for transformer inference with efficient memory scheduling. More specifically, we leverage the one-shot network architecture search method to discover the optimal architecture, where we enlarge the search space by considering low-rank decomposition ratios and token numbers that significantly influence the peak memory during the forward pass. To efficiently acquire effective supernets in the huge search space, we learn the supernet in the fixed search space consisting of given low-rank decomposition ratios and token numbers, and evolve the search space of supernet optimization by predicting the correlation between the task performance and the search space. To construct the operator library within the memory budget, we approximate full-precision operators such as softmax and GeLU layers with integer arithmetics. Moreover, we decompose the patch embedding layers with large convolution kernels to multiple patchification steps with small receptive fields, and overwrite the input token features in the matrix multiplication during the inference for further memory reduction. Extensive experimental results show our method can achieve 73.62% top-1 accuracy on the large-scale image classification dataset ImageNet [11] with the 320KB SRAM STM32F746 microcontrollers, and significantly outperforms the CNN based state-of-the-art networks on MCUs by 5.4%. Our contributions can be summarized as follows:

- To the best of our knowledge, we propose the first hardware-algorithm co-optimizations framework that successfully deploys vision transformers on MCUs with the competitive performance in challenging computer vision tasks

- We present the search space evolution framework for effective architecture search of vision transformers within the memory limits, and construct a operator library with efficient memory scheduling to enable practical deployment.

- We conduct extensive empirical studies to show the feasibility of deploying vision transformers on MCUs with extremely limited memory.

## 2  Related Work

**Efficient vision transformers:** Vision transformers mine the global dependencies via self-attention that outperform convolutional neural networks on a wide variety of tasks such as image classification [22, 37], object detection [5, 40] and instance segmentation [35, 36]. To reduce the computational and storage complexity, efficient architecture design, sparse attention and parameter quantization are presented for efficiency enhancement. For efficient architecture design, Chen *et al.* [6] shared the self-attention in consecutive layers to avoid redundant calculation in dependency discovering, and Fan *et al.* [14] fused multiscale intermediate features to reduce the resolution of top layers. For sparse attention, Rao *et al.* [26] dynamically pruned uninformative tokens during inference according to the computation budget and sample features. For parameter quantization, Liu *et al.* [23] extended the post-training quantization framework by mimicking the self-attention rank in the full-precision model. However, existing methods usually focus on reducing the computational complexity measured by FLOPs and the storage cost evaluated by the number of parameters, and ignore the memory limit in inference that is the main resource bottleneck of MCUs.

**Network architecture search:** Network architecture search aims to discover the topology including operators and connections to achieve the optimal trade-off between the accuracy and complexity, which can be divided into two categories according to the search algorithms. Non-differentiable methods [3, 7, 44] utilize reinforcement learning or evolutionary algorithms to update the selection regarding the defined reward or fitness, where the candidates can be updated from scratch or from a pre-trained supernet. Differentiable methods [21, 41, 42] construct a large supernet that contains all possible choices represented by different parallel branches. By optimizing the importance weights of different choices, the operators and connections with the highest importance weight are selected to form the final architecture. In this paper, we focus on learning the supernet in the optimal search space defined by the low-rank decomposition ratio and token numbers, which influence the inference memory significantly.

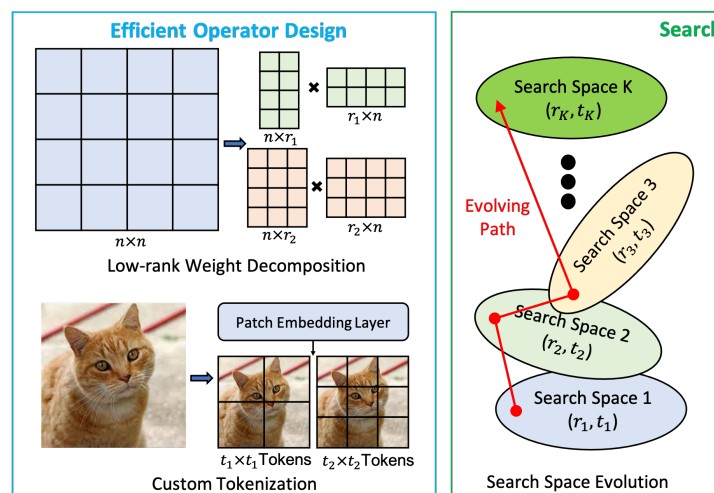

Figure 1: The pipeline of the network architecture search with memory constraint. We search the low-rank decomposition ratio and the token numbers for search space evolution, where the evolutionary process is decided according to regressed tendency between the defined score and the search space factors. With the optimal search space, the architecture with the highest performance within the memory limit is acquired from the trained supernet.

**Deep learning on microcontrollers:** Due to the low price and energy consumption, employing microcontrollers for deep learning inference and training has been comprehensively researched in recent years. Existing frameworks contain TensorFlow Lite Micro [1], CMSIS-NN [18], CMix-NN [4], MicroTVM [10] and TinyEngine [20]. Nevertheless, these compiling libraries are designed for convolutional neural networks and lack operators of vision transformers such as LayerNorm and GeLU. Meanwhile, the layer-wise optimization of memory footprint cannot fully utilize the SRAM budget of the microcontrollers and fails to deploy vision transformers with high task performance. For the memory-efficient network architecture design, Lin *et al.* reduced the image resolution and the width multiplier for peak memory reduction. Sun *et al.* leveraged the mixed-precision quantization that assigned the optimal bitwidth for each layer to achieve the optimal accuracy-memory trade-off, and Saha *et al.* [27] decreased the resolution of the activation after the proposed RNNPool layer. However, the memory reduction techniques are designed for convolutional neural networks, which are not feasible for vision transformers with different operators.

## 3 Approach

In this section, we detail the hardware-algorithm co-optimizations framework to deploy vision transformers on microcontrollers. We first introduce the network architecture search of vision transformers with limited memory constraint, and demonstrate the operator library construction for vision transformer inference on microcontrollers.

### 3.1 Network Architecture Search with Memory Constraint

Conventional architecture search methods usually consider the computational complexity (FLOPs) and the storage cost (the number of parameters), while the peak memory obscures the deployment of vision transformers on microcontrollers due to the extremely limited SRAM. We extend the one-shot NAS to search the optimal architecture, where we first learn some of supernets randomly which containing all possible topology choices and employ evolutionary algorithms to seek for the optimal supernet given the resource budget. Figure 1 demonstrates the pipeline of the network architecture search with memory constraint. To fully leverage the achievements of vision transformer NAS methods, we build the search space of sub-networks on the basis of AutoFormer [8] which searches the embedding dimension, QKV dimension, MLP ratio, head number and depth number with trained hypernetworks. The search space of supernet is defined by the the low-rank decomposition ratio and token numbers that can significantly influence the memory footprint. The token number impacts the size of intermediate features during inference and can be modified by varying the receptive

field in the patch embedding layers. Different from the multiply-accumulate (MAC) operations of convolutional neural networks, the matrix multiplication in large sizes causes heavy memory burden. To address this, we generalize low-rank decomposition for the multi-layer perceptron (MLP) layers in the following form:

$$MLP(\boldsymbol{x}) = \boldsymbol{U}_r \boldsymbol{V}_r^T \boldsymbol{x} + \boldsymbol{b}, \tag{1}$$

where $\boldsymbol{x} \in \mathbb{R}^{n \times d}$ means the input features with $n$ tokens and $d$ elements of each token. $\boldsymbol{U}_r \in \mathbb{R}^{d \times r}$ and $\boldsymbol{V}_r \in \mathbb{R}^{d \times r}$ are two low-rank matrices approximating the original weights in the MLP layer, and $\boldsymbol{b}$ means the bias term. Since the dimension of low-rank matrices affects the trade-off between memory footprint and task performance, we are required to find the optimal low-rank decomposition ratio in our search space selection.

Directly integrating the token numbers and the low-rank decomposition ratio into the original search space in AutoFormer causes convergence difficulties of supernets because of the large discrepancy among optimal architectures of different low-rank decomposition ratios. Therefore, we first optimize the search space of hypernetworks that is composed of the token numbers and the low-rank decomposition ratio, and then performs architecture search in the updated search space of hypernetworks. The above two steps are iteratively implemented until convergence or achieving the maximum search cost. Enumerating all search space for architecture search is not feasible due to the extremely high training cost for a single supernet in the given search space. Inspired by the observation in [9] that there is a strong correlation between the task errors and the continuous choices, we evolve the search space by considering the relation among the performance, memory footprint and the search space factors including token number and the low-rank decomposition ratio. The score function $S$ to evaluate the optimality of different hypernetwork search space is defined as follows:

$$S(A, R) = A + \eta R, \tag{2}$$

where $A$ is the accuracy of the hypernetwork in the search space and $\eta$ is a hyperparameter. $R$ is the ratio of sub-networks whose memory footprint is within the SRAM limit. Meanwhile, enumerating all sub-networks in the supernet to acquire the average memory is prohibitive due to the extremely large numbers of candidates in the subspace, and we randomly sample $N$ sub-networks to approximately evaluate the value of $R$ for the given hypernetwork. The linear fitting functions are employed to estimate the tendency between the score and the factors of the search space as $S = \boldsymbol{w}^T \boldsymbol{x} + S_0$ with the intercept $S_0$, where $\boldsymbol{x} = [x_1, x_2]$ consists of low-rank decomposition ratio and token numbers. The search space is updated with the following rules:

$$x_i^{(t+1)} = x_i^{(t)} - \left[ \frac{w_i}{h_i} \right] \cdot \Delta_i, \quad i = 1, 2, \tag{3}$$

where $x_i^{(t)}$ means the $i_{th}$ element in $\boldsymbol{x}$ at the $t_{th}$ step and $w_i$ represents the $i_{th}$ element in $\boldsymbol{w}$. The optimization step $\Delta_i$ is set to the interval of adjacent factor values in our method, and the update threshold $h_i$ are hyperparameters for the optimization. $[x]$ demonstrates the largest integer smaller than $x$.

Training the hypernetwork in the evolving search space until convergence requires unacceptable optimization cost, and we fit the correlation between task performance and the continuous choices of search space factors in the training process. For each evolved search space, we train the supernet for $t$ epochs from the resume points or scratch if pre-trained model does not exist. To fairly discover the tendency between the task performance and the continuous choices, we fit the correlation weight matrix $\boldsymbol{w}$ with the data points in the same total training epochs. Since the linear fitting function only holds in a small local region for a given search space, we utilize the piecewise linear fitting function that only considers top-k nearest neighbors for the search space factor $\boldsymbol{x}$. The data point sampling strategy for tendency fitting can be represented as follows:

$$\Phi(\boldsymbol{x}) = \{\boldsymbol{z} | \boldsymbol{z} \in \mathcal{N}_k(\boldsymbol{x}), E(\boldsymbol{z}) = E(\boldsymbol{x})\}, \tag{4}$$

where $\Phi(\boldsymbol{x})$ means the set of data points consisting of task accuracy with memory and continuous choices of low-rank decomposition ratio and token numbers for fitting the linear function in the evolving search space $\boldsymbol{x}$. The neighborhood for $\boldsymbol{x}$ with $k$ nearest neighbors is denoted as $\mathcal{N}_k(\boldsymbol{x})$, and the total training epochs for the supernet in the search space $\boldsymbol{x}$ are represented by $E(\boldsymbol{x})$. When the neighborhood of the given search space $\boldsymbol{x}$ is empty, we construct the neighborhood by considering the top-k nearest neighbors whose total training epochs are $E(\boldsymbol{x}) - t$. We train hypernetworks in

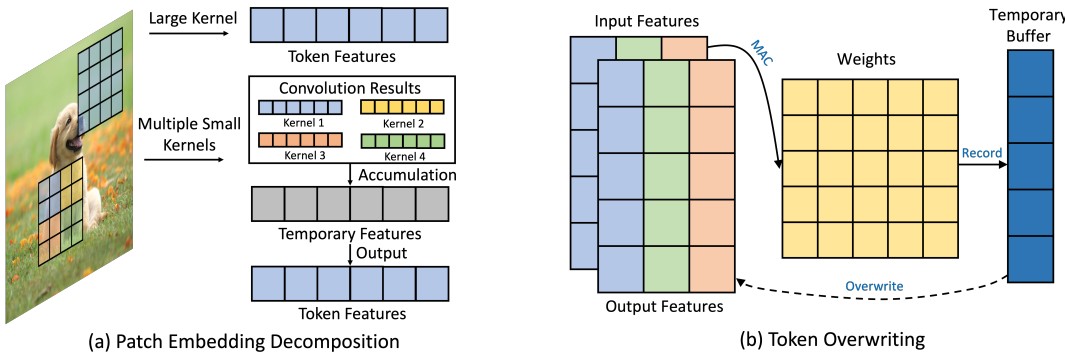


Large Kernel → Token Features

Convolution Results
Kernel 1   Kernel 2
Kernel 3   Kernel 4
↓ Accumulation
Temporary Features
↓ Output
Token Features

Multiple Small Kernels

(a) Patch Embedding Decomposition

Input Features — MAC — Weights — Record → Temporary Buffer
Output Features — Overwrite

(b) Token Overwriting


Figure 2: (a) The comparison between the conventional patch embedding operation and our decomposition. We decompose the convolution operator with a large receptive field into multiple operators with small receptive fields and utilize a temporary buffer to record the intermediate accumulations. (b) Demonstration of token overwriting. The buffer for input features is re-used by the output features, and the peak memory is reduced significantly.

these search space for $t$ epochs for accuracy update to fairly estimate the correlation matrix $w$ for the given search space $x$.

The selected supernet in the optimal search space is used for subsequent training until full convergence and is employed for the network architecture search of vision transformers. In this process, the evolutionary algorithm is applied with the aim of maximizing accuracy while adhering to the memory constraint.

## 3.2 Operator Library Construction for Vision Transformers

Deploying the vision transformer on microcontrollers for inference requires the operator library to transform the model into executable functions. Existing operator libraries for deep learning are comparable with convolutional neural networks, while fail to implement operations in vision transformers such as the GeLU activation function and the layer normalization. Meanwhile, the inference efficiency of the library cannot fully utilize the memory budget of microcontrollers to achieve higher task performance with larger vision transformer models. Despite of the efficient memory scheduling techniques that are general to all network architectures in [20], we present the following memory utilization enhancement techniques to enable deployment of vision transformers on microcontrollers.

**Patch embedding decomposition:** Images are patchified by the patch embedding layer in vision transformers for inference. The input samples are transformed into patches by convolution operations, where both the kernel size and stride are both equal to the patch size. Since the kernel size in the patch embedding layer (larger than $16 \times 16$) is much larger than common convolutional neural networks, most memory cost of the patch embedding layer results from the filter weights instead of the input and output activation. In order to effectively address this challenge, we decompose the patch embedding layer by transforming the convolution with large receptive field to multiple convolution operations with a small receptive field. To add the results from multiple convolution operations, we leverage a small memory buffer to record the intermediate value of accumulation, effectively reducing the peak memory usage during image patchification. Figure 2a demonstrates the comparison between the original patch embedding operation and that with our decomposition. As decomposing the patch embedding layer into excess convolutional operators with small receptive fields can obviously increase the inference latency due to the multiple forward passes, we select the receptive field of the decomposed patch embedding layer as $4 \times 4$ to achieve the satisfying trade-off between the memory footprint and the inference latency.

**Operator integration:** Quantizing network parameters and substituting MAC operations with integer arithmetics have been proven beneficial in reducing the memory of convolutional neural networks without causing performance degradation. However, conventional operator library with quantized operators is not feasible for vision transformer inference because of the specific operators including the GeLU activation and layer normalization. Since GeLU requires the cumulative distribution

function (CDF) of Gaussian distribution, we approximate the activation function with quantization by multiplication and softmax function which can both implemented with int8 arithmetics. Similarly, the square root operators in the layer normalization is not supported by int8 arithmetics in operator library. We construct surrogate equations with fixed-point iterative methods to calculate the output of the square root operators. Considering the inference latency of layer normalization operator, we only iterate the surrogate equation for 3 times with balance between the inference latency and the prediction precision. We detail the pseudo algorithm of integer-only square root operation in the supplementary material.

**Token overwriting:** The matrix multiplication in vision transformer inference that results in high memory consumption impose element interaction only within tokens, and the interaction among tokens is enabled in self-attention computation whose memory cost is far less than the SRAM limit. Therefore, we can overwrite the intermediate features during inference of fully-connected layers because the input activation of each token will no longer be used after acquiring the output features of the corresponding token. Figure 2b illustrates the memory scheduling of matrix multiplication in existing inference operator libraries and our token overwriting techniques, where our method writes the output of matrix multiplication for each token in the same memory space as the input activation. Since the peak memory usage occurs in the matrix multiplications, we assign the maximum memory size $M$ for vision transformer inference based on the network architecture, according to the following criteria:

$$M = \sup_l \ h_f^l w_f^l + \max(h_i^l w_i^l, \ h_o^l w_o^l), \tag{5}$$

where $h_f^l$, $h_i^l$ and $h_o^l$ represent the height of weights, input features, output features in the $l_{th}$ fully-connected layers, and $w_f^l$, $w_i^l$ and $w_o^l$ demonstrate the width of the above tensors. Since the memory consumed by the input activation can be overwritten by the output features, we only assign the memory buffer with the same size as the larger one between input and output tensors.

## 4 Experiments

In this section, we first introduce our hardware configuration, dataset and the implementation details of our hardware-algorithm co-optimizations framework, and then we conduct thorough performance analysis including the task performance and the memory consumption with respect to the presented techniques in network architecture search and operator library construction. Finally, we compare the performance in accuracy and efficiency with existing network architecture search methods and operator libraries for deep model deployment on microcontrollers.

### 4.1 Setups

**Hardware configuration:** We deploy the vision transformers with our hardware-algorithm co-optimizations framework in different microcontrollers with various resource constraint including STM32F427 (Cortex-M4/256KB memory/1MB flash), STM32F746 (Cortex-M7/320KB memory/1MB flash) and STM32H743 (Cortex-M7/512KB memory/2MB flash). In the performance analysis, we evaluate our framework on STM32F746 to acquire the accuracy and memory.

**Dataset:** We conduct the experiments on ImageNet for image classification, which contains 1.2 million training images and 50k validation images from 1000 classes. All images are scaled and biased into the range [-1,1] for normalization. For the training process, we resize the images with the shorter side as 256 and randomly crop a 240×240 region. For inference, we utilize the central crop with the size of 240×240.

**Implementation details:** For the network architecture search of vision transformers, our choices for the search space consisting of low-rank decomposition ratio $r$ and the token size $c$ can be selected from $r \in [0.4 : 0.05 : 0.95]$ and $c \in \{16, 20, 24, 28, 32\}$. The supernet design and the evolutionary search in the learned hypernetwork keep the same with those in [8] to fully leverage the potential in architecture search with high degrees of freedom. During the search space evolution, we only select 5 nearest neighbors with the same total training epochs to fit the piecewise linear function for acquiring the slope. For each time of evolving, the supernet in the selected search space is trained for 30 epochs from the resumed points if existed or from scratch. If the neighborhood set for learning the tendency function is empty, we select the top-5 nearest neighbors whose training epochs are 30 less than the given hypernetwork. Then we train hypernetworks in these search space to acquire the

Table 1: Comparison of flash, memory and top-1 accuracy variation across different search space selection method.

| Search Space | Flash | Memory | Top-1 |
|---|---|---|---|
| Random | 1.03MB | 332kB | 65.2% |
| Maximal | 1.11MB | 271kB | 69.2% |
| Composition | 0.96MB | 218kB | 68.6% |
| Ours | 0.89MB | 218kB | 71.1% |

Table 2: The effects of different neighborhood sizes in tendency fitting, where we report the inference efficiency and the top-1 accuracy.

| Neighborhood | Flash | Memory | Top-1 |
|---|---|---|---|
| 2 | 0.79MB | 207kB | 65.4% |
| 3 | 0.84MB | 231kB | 68.2% |
| 5 | 0.89MB | 218kB | 71.1% |
| 8 | 0.89MB | 218kB | 71.1% |

| MCUFormer | Search Space Evolution | Token overwriting | Patch Embedding decomposition | Operator integerization |
|---|---|---|---|---|
| 218kB | 111kB | 155kB | 327kB | 2400kB |

Figure 3: The peak memory reduction is achieved through the operator library with memory scheduling techniques, including patch embedding decomposition, Operator integration, and token overwriting. The peak memory of the final model adheres to the SRAM constraint, which is 256kB.

accuracy for tendency fitting. The initialized data points for tendency fitting are randomly sampled continuous choices. For operator library construction, we utilize int8 quantization for all tensors in the vision transformer during inference. The filter size of the decomposed patch embedding layer is set to $4 \times 4$ with multiple forward passes to reduce the peak memory, and we iterate the surrogate assignment from the fixed-point iterative methods for 4 times to calculate the square root in the layer normalization operators. The hyperparameter assignment and approximation method is listed in the supplementary material.

## 4.2 Performance Analysis

**Effects of search space evolution:** Low-rank decomposition ratio and token number influence the memory significantly for vision transformer deployment. In order to discover the optimal search space with satisfying trade-off between task performance and memory, we evolve the search space efficiently according to the estimated tendency. We compare our search space evolution method with other selection criteria including randomly sampled search space, the maximal search space and composing the search space selection into the supernet learning. Table 1 shows the top-1 accuracy, the memory footprint and the flash of different methods, where our approach outperforms others in both the accuracy and efficiency. Randomly selecting the search space fails to discover the optimal trade-off between the task performance and the efficiency of storage and memory, while choosing the search space with the highest complexity underperforms the evolution methods because of the low ratio of sub-networks within the memory limit. Di-

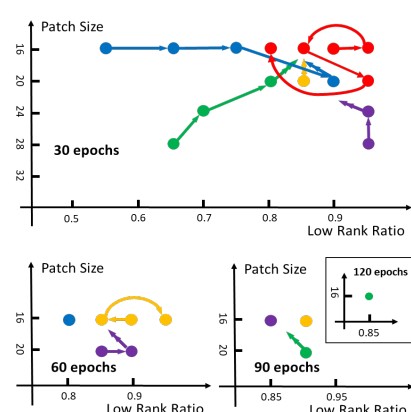

Figure 4: Visualization of search space evolution.

rectly composing the search space selection in supernet training amplifies the search complexity by 15-16 orders of magnitude, and the search deficiency prevents the acquisition of the optimal network architectures.

**Effects of memory scheduling in the operator library:** To enforce the vision transformers to be executable on microcontrollers, we construct an operator library to include the lack functions required by transformers. Moreover, we also reschedule the memory during inference to fully utilize the memory constraints of microcontrollers through patch embedding decomposition, Operator integration and token overwriting. Figure 3 illustrates the comparison of peak memory for the operator library with different memory scheduling techniques. Operator integration makes the most significant contribution to memory reduction (23.8%) because the memory to represent the int8 tensor only costs 25% of that for a float one. Patch embedding decomposition is necessary because the large

convolutional kernels requires large memory space for the MAC operation in convolution. Token overwriting also decreases the memory cost by reusing the buffer for each step of multiplication, since the memory for both input and output can be replaced by that only represents the larger one of them. In conclusion, integrating all above techniques into the operator library reduces the memory cost by 9% and enables deployment of large-scale vision transformers on microcontrollers with satisfying task performance.

**Influence of tendency fitting:** The search space evolves according to the fitted tendency between the continuous choices and the task performance and memory, and effective tendency estimation is important to acquiring the optimal network architectures. Considering both the simplification and the fitting precision, we leverage the piecewise linear function to fit the tendency with the neighboring data points near the given search space. Large neighborhoods are more robust to outliers of task performance and continuous choices, while small neighborhoods generate more precise fitting functions for the selected search space. We investigate the influences of the neighborhood size on tendency fitting, and Table 2 depicts the results for data point sampling with different numbers of nearest neighbors. The assignment of moderate numbers of data points in the neighborhood results in the architecture with the highest accuracy within the memory constraints due to effective search space evolution. We also provide a visualization of a search space evolution example in Figure 4, where the search space that is selected more frequently during evolution is verified to be more optimal. Different colors represent the evolutionary paths of various candidates, and the arrows indicate the direction of evolution. Double arrows signify that the candidates after evolution are trained with $t$ more times than current total training epochs.

**Influence of hyperparameters in search space evolution:** Crucial hyperparameters in search space evolution contains the update threshold $h_1$ and $h_2$, which are varied in the ablation study to investigate the influence. Figure 5 demonstrates the top-1 accuracy for different value assignment of thresholds, where both medium thresholds result in the highest performance. Large thresholds cause frequent search space change without sufficient confidence, while the small ones enforce the evolution process to stuck at the local optimum. Meanwhile, the performance is more sensitive to the update threshold for low-rank decomposition ratio, which significantly changes the optimal architecture.

## 4.3 Evaluation on Large-scale Image Classification

In this section, we compare the network architecture search techniques in MCUFormer with state-of-the-art architecture design methods including the vision transformer based NAS methods including AutoFormer [8].We also compare our operator library with existing ones designed for convolutional neural network inference. Since the conventional operator library such as CMSIS-NN [18] and TinyEngine [20] cannot be directly utilized for vision transformer inference due to the lack of executable functions for specific design such as GeLU and layer normalization, we supplemented the missing functions to the conventional library to ensure a fair comparison of memory scheduling in vision transformer inference.

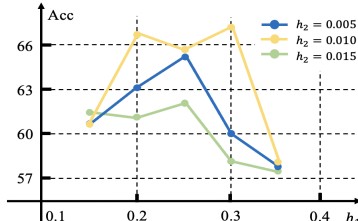

Figure 5: The performance variation with respect to the threshold $h_1$ and $h_2$ in search space evolution.

Image classification on ImageNet is highly challenging, primarily because of its large scale and high diversity, particularly when using lightweight models deployed on microcontrollers. Table 3 illustrates the top-1 classification accuracy, the memory footprint and flash of different deep neural networks including CNN based and transformer based architectures and various operator libraries. The gray number in the table indicates the the required memory or flash exceeds the memory constraints or flash constraints. For the combination of architectures and operator libraries that exceeds the memory constraint of the microcontrollers, we report the accuracy achieved during inference with GPUs. For the CNN based architectures[28, 30], we employ the TinyEngine [20] for compilation. MCUFormer-A and MCUFormer-T respectively represent our method that replaces the network architecture search with AutoFormer and utilizes the TinyEngine with supplemented layers of vision transformers for compilation. Except for MCUFormer-T, we employ our proposed engine for compilation to evaluate all transformer based architectures.

Since vision transformers outperform CNNs by a sizable margin on large-scale vision tasks, deploying transformers on microcontrollers is very desirable for practical applications. Compared with the

Table 3: The flash, memory and top-1 accuracy on ImageNet for different network architecture methods and deep learning libraries for microcontrollers, where we utilize three MCUs with various SRAM limits for evaluation.

| Model | STM32F4 256KB | | | STM32F7 320KB | | | STM32H7 512KB | | |
|---|---|---|---|---|---|---|---|---|---|
| | Mem. | Fla. | Acc. | Mem. | Fla. | Acc. | Mem. | Fla. | Acc. |
| MBV2+CMSIS | - | - | - | 308kB | 0.72MB | 49.0% | - | - | - |
| MCUNetV1 | 238kB | 0.70MB | 60.3% | 293kB | 0.70MB | 61.8% | 452kB | 1.65MB | 68.5% |
| MCUNetV2 | 233kB | 0.67MB | 62.0% | 282kB | 0.67MB | 63.5% | 498kB | 1.56MB | 70.7% |
| EMQ | 253kB | 0.73MB | 66.5% | 308kB | 0.71MB | 68.2% | 507kB | 1.67MB | 72.8% |
| AutoFormer | - | - | - | 681kB | 1.24MB | 74.7% | - | - | - |
| MCUFormer-A | 328kB | 0.86MB | 70.4% | 411kB | 0.98MB | 69.3% | 842kB | 2.11MB | 72.7% |
| MCUFormer-T | 816kB | 3.81MB | 73.3% | 872kB | 3.90MB | 75.4% | 1.5MB | 7.45MB | 76.4% |
| MCUFormer | 218kB | 0.89MB | 71.1% | 319kB | 0.90MB | 73.6% | 505kB | 1.95MB | 74.0% |

state-of-the-art CNN based model EMQ on microcontrollers, our method achieves 5.4% higher top-1 accuracy (68.2% vs. 73.6%) with the 320KB memory limit. Both the search space evolution in architecture search and the memory scheduling in the constructed operator library contribute to deploying vision transformers on microcontrollers with high accuracy by comparing MCUFormer-T, MCUFormer-A and MCUFormer, because the redundant representation that consumes the memory of MCUs in the inference is removed. Although the state-of-the art NAS methods AutoFormer [8] effectively discover the topology of vision transformers with the optimal accuracy-complexity trade-offs, they fail to consider the extremely memory constraints of microcontrollers which are not a concern for GPUs. Directly supplementing the low-rank decomposition ratio and token numbers to the original search space usually causes search deficiency because of the convergence problems in the supernet training. Meanwhile, TinyEngine optimizes the memory scheduling for general deep network deployment while still ignores the specific architectures in vision transformers for further memory utilization ratio enhancement. Generally speaking, the presented search space evolution in network architecture search and memory scheduling in library construction respectively boost the top-1 accuracy by 4.3% (69.3% vs. 73.6%) and reduce the peak memory by 63% (319kB vs. 872kB) with the STM32F746 which get better result.

We also deploy our hardware-algorithm co-optimizations framework in different controllers with various resource constraint, and Table 3 also demonstrates the top-1 accuracy for various methods. With 256KB memory and 1MB flash, our vision transformer still achieves 71.1% top-1 accuracy on ImageNet and outperforms the EMQ method by 6.9%. The task performance is boosted with the increase of the constraint of memory and flash, where the acquired the 74.0% top-1 accuracy on STM32H743 with 512KB memory and 2MB flash even outperforms the Deit [32] 1.8% inferenced on GPUs. Hence, our vision transformer can be applied in practical applications with high requirement of accuracy.

## 5 Conclusion

In this paper, we have presented MCUFormer for hardware-algorithm co-optimizations that successfully deploys the vision transformer on microcontrollers with satisfying task performance. We generalize the network architecture search with enlarged search space considering the low-rank decomposition ratio of weight matrix and token number. The search space evolves for supernet training to discover the model topology with the highest accuracy given the resource constraints, where the evolution leverages the tendency between task performance and the continuous choices with enhanced search efficiency. Meanwhile, we construct the operator library for vision transformer inference that converts the model into executable functions, and fully utilize the memory by patch embedding decomposition, Operator integration and token overwriting. Extensive experiments on image classification, person presence identification and key word spotting shows that our MCUFormer achieves competitive performance on microcontrollers with the state-of-the-art vision transformer models. Our work currently contains the following limitation. Considering transformer deployment on MCUs in more diverse vision tasks such as DETR [5] for object detection is important in realistic applications including autonomous driving[38, 25] and robot navigation[15, 13].

## Broader Impacts

Deploying transformer on a microcontroller is particularly beneficial for applications that require quick decision-making or operate in resource-constrained environments. Unlike the traditional CNN, the peak memory of the transformer will not change with the number of channels, so it is relatively difficult to reduce the peak memory. The MCUFormer we designed reduces the peak memory from the structural design and improves the accuracy at the same time.

## Acknowledgments

This work was supported in part by the National Key Research and Development Program of China under Grant 2022ZD0160102, and in part by the National Natural Science Foundation of China under Grant 62321005, Grant 62336004, and Grant 62125603.

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
