# A Operator integration

Current operator library with quantized operators is not feasible for vision transformer inference because of the specific operators including the GeLU activation and layer normalization. Layer normalization (LayerNorm) normalizes the activations of each layer in a neural network independently, reducing internal covariate shift and improving training stability as follows:

$$\text{LayerNorm}(x) = \frac{\gamma}{\sqrt{\text{Var}(x) + \epsilon}} \cdot (x - \mu) + \beta, \tag{1}$$

where $x$ is the input tensor. $\mu$ is the mean of the input tensor, and $Var(x)$ is the variance of the input tensor.The square root operators in the layer normalization is not supported by int8 arithmetics in operator library. We construct surrogate equations with fixed-point interactive methods to calculate the output of the square root operators inspired by I-BERT[3]. We provide the details of how to approximate the square root operators in Algorithm.1. GeLU requires the cumulative distribution function (CDF) of Gaussian distribution, we approximate the activation function by Equation.2[1].

$$\text{GeLU}(x) = \sigma(1.702x), \tag{2}$$

---

**Algorithm 1:** Integer-only Square Root Operation

---

**input** : $x \geq 0$
**output** : $\lfloor y = \sqrt{x} \rfloor$
**if** $x = 0$ **then**
  |   $y = 0$
**else**
    Intialize $P = \lceil \log_2 x \rceil$, $x_0 = 2^{\lceil P/2 \rceil}$, $i = 0$
    **repeat**
      |   $x_{i+1} = \lfloor (x_i + \lfloor x/x_i \rfloor)/2 \rfloor$ $i = i + 1$
    **until** $i = t$;
**end**

---

# B Experiment Details

## B.1 Evolution Pipeline

To find the optimal search space, we evolve the search space by considering the relation among the performance, memory footprint and the search space factors. Algorithm.2 shows the evolution search in our method.

---

**Algorithm 2:** Evolution Pipeline

---

**input** : search space $S$, candidate search spaces $S_1, ..., S_n$, the number of candidate search
        spaces $n$, iteration $\tau$, the number of sample architectures $N_{acc}, N_{ratio}$
**output** : the optimal search space $S_{optimal}$
Randomly choose n search spaces as candidate search spaces
**for** $i \leftarrow 0$ **to** $\tau$ **do**
    Train $S_1^{(i)}, ..., S_n^{(i)}$ for $t$ epochs and get $S_1^{(i+1)}, ..., S_n^{(i+1)}$
    Randomly sample $N_{acc}$ architectures from $S_1^{(i+1)}, ..., S_n^{(i+1)}$ to get the accuracy $A$
    Randomly sample $N_{ratio}$ architectures from $S_1^{(i+1)}, ..., S_n^{(i+1)}$ to get the ratio $\eta$
    Calculate the score function according to equation 2 in the paper
    Evolve the search space according to equation 3 in the paper
**end**
The optimal search space $S_{optimal}$ is selected more times
Train the optimal search space $S_{optimal}$

---

## B.2 Hypernetwork Search Space

We set hypernetwork search space with the following factors.

1. Embedding Dimension for the whole network, choosing from [168 : 24 : 216]
2. MLP ratio for MLP block, choosing from [3.5 : 0.5 : 4.0]
3. Depth for layer number, choosing from [12 : 1 : 14]
4. Head Numbers for multi-head attention block, choosing from [3 : 1 : 4]
5. low-rank decomposition ratio for matrix multiplication, choosing from [0.4 : 0.05 : 0.95]
6. token size for the whole network, choosing from [16 : 4 : 32]

## B.3    Training Details

We train the hypernetwork using a similar recipe as AutoFormer[2].The training dataset is ImageNet and the validation dataset subsample 10,000 images from ImageNet. Data augmentation techniques, including RandAugment, Mixup and random erasing, are adopted with the same hyperparameters as in DeiT[4].

## B.4    Evolutionary search

We used evolutionary search as AutoFormer[2] to find the best sub-network architecture under certain constraints. We use a population size of 50. We randomly sample 50 sub-networks satisfying the constraints. We perform crossover to generate 25 new candidates and mutation to generate another 25, forming a new generation. The mutation probability $P_d$ and $P_m$ are set to 0.2 and 0.4 We repeat the process for 20 iterations and choose the sub-network with the highest accuracy on the split validation set.