# OpenReview forum: "MCUFormer: Deploying Vision Tranformers on Microcontrollers with Limited Memory"
_NeurIPS.cc/2023/Conference — NeurIPS 2023 poster_

### Official Review · Reviewer_9JB9 · 2023-06-27

**Soundness:** 3 good
**Presentation:** 2 fair
**Contribution:** 2 fair
**Rating:** 5
**Confidence:** 4

**Summary:**

This paper focuses on the efficient deployment of vision transformers on microcontroller platforms with limited resources. Vision transformers face significant challenges when deployed on resource constrained MCUs due to their large volume of data movement and intensive computation size. The paper explores algorithmic and systematic optimizations by modifying model architectures and reducing memory footprints. In summary, this paper proposes a hardware-algorithm co-design method called MCUFormer that enables the deployment of vision transformers on microcontrollers. This is achieved through joint design of transformer architectures and construction of an inference compiler that fits within the memory constraints.

**Strengths:**

- This work targets an important research problem, and the proposed method generally makes sense. The evaluation shows promising results in terms of accuracy and memory consumption.
- The proposed method is extensively evaluated on three kinds of resource constraint platforms and an ImageNet dataset.
- The discussion on the breakdown of memory reduction is good.

**Weaknesses:**

- Some helpful details regarding speed comparison are missing, which would be useful in demonstrating the performance gain against other optimization methods and existing work.
- The paper claims to have a hardware-algorithm co-design, but it is unclear where the Hardware design exists. It is more like the co-optimizations between algorithm and compiling system.
- Existing work has widely discussed NAS with different constraints (e.g., latency, memory footprint, energy), so it would be helpful to further clarify the differences among them.
- Regarding systematic optimizations, there seems to be limited novelty. Efficient quantized operator or integer-only inference implementation has already been discussed in existing work, e.g., [1-3]. It appears that operator integration provides another implementation for specific quantized operators/kernels. Temporary buffer reusing is also widely used in existing inference frameworks such as TFLite and MNN.
- Minor issues include improving the writing style and some typos: "Operator integerization" to "Operator integration."

[1] A Microcontroller is All You Need: Enabling Transformer Execution on Low-Power IoT Endnodes

[2] FQ-ViT: Post-Training Quantization for Fully Quantized Vision Transformer

[3] I-ViT: Integer-only Quantization for Efficient Vision Transformer Inference

**Questions:**

Please see above for detailed comments. Additionally, I have some questions:

- The paper evaluates the framework on STM32F746 to measure accuracy, latency, and memory usage. However, I cannot find any information regarding latency or speedup results in the main paper. Has this been discussed elsewhere?
- It would be beneficial if the author could clarify whether the accuracy data was collected from a desktop platform or an MCU device.

**Limitations:**

Yes

---

> ### Author Rebuttal · Authors · 2023-08-10
>
> Thank you for careful reading and valuable comments. We will check the paper carefully, and modify the presentation of the ambiguous parts in the final version. We provide answers to the questions as follows:
>
> Q1: Compare with existing work.
>
> A1: Due to lack of operator library for vision transformer inference, we cannot deploy transformer on MCU based on existing NAS work. So we compared with CNN based solution. Since the article only trains the model on GPU and do not deploys the model on GPU, we can not get the latency. We will compared with MCUNet in the following table. All the latency is normalized to STM32F7. Compared with CNN based model, MCUFormer archieves higher accuracy and have the similiar latency.
>
> |           | Peak Memory | Accuracy | Latency |
> |-----------|-------------|----------|---------|
> | MCUNetV1  | 293kB       | 61.8%    | 890ms   |
> | MCUNetV2  | 282kB       | 63.5%    | 501ms   |
> | MCUFormer | 319kB       | 73.6%    | 641ms   |
>
> Q2: The results of inference latency
>
> A2: We deploy vision transformers using our algorithm-hardware framework across diverse microcontrollers characterized by varying resource constraints. Our assessment of the framework encompasses a comprehensive evaluation, encompassing accuracy, latency, and memory usage. Our method achieves higher top-1 accuracy and fit the memory constraint.
>
> | MCU      | STM32F4 | STM32F7 | STM32H7 |
> |------------|----------|---------|---------|
> | Constrain  | 256kB    | 320kB   | 512kB   |
> | Accuracy   | 71.1%    | 73.6%   | 74.0%   |
> | Latency    | 706ms    | 641ms   | 798ms   |
> | Memory     |  218kB   | 319kB   | 505kB   |
>
> Q3: co-design system
>
> A3: We are sorry for the ambiguity. For algorithm, We first optimize the search space of hypernetworks that is composed of the token numbers and the low-rank decomposition ratio, and then performs architecture search in the updated search space of hypernetworks. The evolution algorithm gets the balance between accuracy and memory constrains. For hardware design, we extend the operator library, use overwriting to share buffer and patch embedding decomposition to reduce the memory. Initially, co-design means both the algorithm and hardware design reduce the memory to fit the constraints on MCU. We will replace the co-design by co-optimizations in the final version.
>
> Q4: Limited novelty
>
> A4: Compared with [4], we have less error due to less approximation rewrite. And we have the co-optimization system to reduce the memory. Compared with [5][6], we construct a compiling library with efficient memory scheduling to enable practical deployment.
>
> (1)	efficient quantized operator or integer-only inference
>
> Firstly, compared with[4], they approximate softmax operator which will influence the accuracy. To get the balance between the inference latency and the prediction precision, we only make Layernorm and GeLU approximation and we achieve 8-bit inference on GPU.
> Secondly, Since we should use existed operators to translate model and the translation of layers influences latency, FQ-ViT[2] and I-ViT[3] cannot get better performance when deploying on MCU. During the process of inference, [5][6] use the power function, which will greatly increase the latency, and also contain operators that cannot be translated. But for MCUFormer, we construct surrogate equations with fixed-point iterative methods for 3 times to calculate the output of the square root operators which can be translated.
>
> (2)	existing inference frameworks
>
> Though it is easy to support cross-platform development, TF-Lite requires extra runtime memory, to store the meta-information (such as model structure parameters). For overwriting, TF-Lite will use additional temporary buffer which will increase the peak memory. Our method writes the output of matrix multiplication for each token in the same memory space of the input activations and share the same memory.
>
> Q5: Typos
>
> A5: We are sorry for typos, and we will make the correction and replace ‘Operator integerization’ by ‘Operator integration’ in the final version. Thank you for pointing it out.
>
> Q6: Accuracy data collection
>
> A6: Following the settings in MCUNet and EMQ, we report the accuracy acquired on inference with GPUs.
>
> [1] MCUNet: Tiny Deep Learning on IoT Devices
> [2] Entropy-Driven Mixed-Precision Quantization for Deep Network Design
> [4] A Microcontroller is All You Need: Enabling Transformer Execution on Low-Power IoT Endnodes
> [5] FQ-ViT: Post-Training Quantization for Fully Quantized Vision Transformer
> [6] I-ViT: Integer-only Quantization for Efficient Vision Transformer Inference

---

> > ### Comment · Reviewer_9JB9 · 2023-08-19
> >
> > I appreciate the author's detailed responses and data provided for my queries, which partially solved my concerns. I've adjusted the rating accordingly.

---

> > > ### Author Response · Authors · 2023-08-20
> > > **Response to reviewer 9JB9**
> > >
> > > We extend our heartfelt gratitude for your positive evaluation of our paper. Your valuable insights are greatly appreciated, and we are committed to incorporating your feedback to enhance the quality of our paper.

---

### Official Review · Reviewer_77fa · 2023-06-30

**Soundness:** 3 good
**Presentation:** 3 good
**Contribution:** 2 fair
**Rating:** 5
**Confidence:** 4

**Summary:**

This paper introduces MCUFormer, which aims to deploy vision transformers on micrcontrollers. MCUFormer utilizes two key techniques: 1. search space evolution to search vision transformer architectures within the memory limits; and 2. the construction of a compiling library with efficient memory scheduling for deployment on real-world microcontrollers. MCUFormer achieves 73.62% top-1 accuracy with only 329KB of memory.

**Strengths:**

1. The paper is well-written and easy to follow
2. The compiling library, which incorporates several memory optimization techniques, is both effective and novel
3. The final achieved performance is competitive.

**Weaknesses:**

1. The proposed search space evolution algorithm appears to be an incremental improvement compared to the work[1].
2. The proposed search space evolution algorithm is quite complex, and Figure 3 suggests that it may not be the most significant contributor to reducing memory footprint. It is not clear why simpler methods, such as model pruning followed by token pruning, were not used to compress a pre-trained vision transformer.
3. The paper claims to have a system-algorithm co-design approach. However, the two main techniques proposed in the paper seem to be independent (i.e.: the first one is for algorithm and the second is for system optimization). It is not clear how these two techniques are integrated in a co-design approach.
4. Some key details are missing. For example, the total search cost of the proposed search space evolution algorithm; the memory constraints used during the search, and the final architecture search processes, etc.


[1] Chen, Minghao, et al. "Searching the search space of vision transformer." Advances in Neural Information Processing Systems 34 (2021): 8714-8726.

**Questions:**

1. The paper mentions that supplementary materials can be referenced, but it seems that the authors may have forgotten to upload them.
2. Can you provide more details about your proposed search space evolution algorithm? For example, what is the total search cost, and what memory constraints were used? How did you evaluate the memory consumption of candidate sub-networks during the search? This is a crucial setting that seems to have been missed.
3. Can you comment on the comparison between using model compression and token pruning algorithms on pre-trained transformers and your proposed search space evolution algorithm?

I will do the final decision based on your rebuttal answers.

**Limitations:**

The proposed search space evolution algorithm appears to be incremental and the paper’s claim of a system-algorithm co-design approach is unclear. The algorithm is complex and may not significantly reduce memory footprint.
I'm not sure the novelty of the techniques used in the compiling library is enough.

---

> ### Author Rebuttal · Authors · 2023-08-10
>
> Thank you for careful reading and valuable comments. We will check the paper carefully, and modify the presentation of the ambiguous parts in the final version. We provide answers to the questions as follows:
>
> Q1: Difference between AutoFormerV2[1] and MCUFormer
>
> A1: Traditional vision transformers including ViT, AutoFormer, Q-ViT are trained or inferenced on GPU. MCUFormer inferences on MCU which SRAM and flash are much lower than CPU. Compared with AutoFormerV2, our method uses the peak memory as the limitation instead of parameter. Our main contribution is letting the vision transformer model meet memory constraint.
>
> (1)	search space enlargement
>
> Existing transformer model cannot deploy on MCUs due to the memory constraint. AutoFormerV2 finds the optimal window size, but larger embedding kernel size will increase the buffer usage. Since the dimension of located tensor affects peak memory[2], low-rank decomposition reduces the large matrix dimensions and lower token number reduces the size of intermediate features, we compose two factors into initial search space to find optimal search space.
>
> (2)	search space decomposition.
>
> Composed with low-rank ratio and token number, hypernetwork contains more than 8×10¬¬30 candidate architectures. Due to the convergence difficulties of hypernetwork composited with two factors, the accuracy of network is much lower. Therefore, we optimize the search space of hypernetworks that are composed of the token numbers and the low-rank decomposition ratio. For the detail information of other selection criteria, you can refer to the Table1 in article.
>
> (3)	evolution mode difference.
>
> AutoFormerV2 also proposes automatic search space design method which could modify the number of blocks contained in stage. But these cannot change the peak memory. For MCUFormer, enumerating all search space for architecture search is not feasible due to the extremely high training cost for one hypernetwork in the given search space. We perform architecture search in the updated search space of hypernetworks. Since two factors influence the trade-off between accuracy and memory constrains, we fit the correlation weight by current accuracy of hypernetwork and the ratio of eligible subnets whose memory footprint is within the SRAM limit. Moreover, We train hypernetworks in these search space for limited epochs for accuracy update to fairly estimate the correlation matrix for the given search space.
>
> Due to AutFormerV2[1] based on swin transformer, we compared with AutoFormer, AutoFormer composed with two factors and MCUFormer on STM32F4 which could refer to following table.
>
> |           | Accuracy | Peak Memory | GPU Hours |
> |-----------|----------|-------------|-----------|
> | AutoFormer| 73.7     | 1.24MB      | 200       |
> | Composed  | 68.6     | 218kB       | 600       |
> | MCUFormer | 71.1     | 218kB       | 360       |
>
> Q2: Effect of search space evolution algorithm in the whole deploying process
>
> A2: For STM32F4, which memory constraint is 256kB, the search space evolution algorithm reduces the peak memory from 329kB to 218kB. We add some ablation study in the following table to demonstrate the effect of search algorithms and hard ware design. Moreover, fixed low-rank ratio and token number cannot get the optimal search space optimal architectures.
>
> |                       | Memory  |
> |-----------------------|---------|
> | ViT + quant           | 811kB   |
> | ViT + quant + evolution| 487kB   |
> | ViT + quant + hardware design| 329kB |
> | MCUFormer             | 218kB   |
>
>
> Q3: Difference between search space evolution algorithm and model compression and token pruning algorithms
>
> A3: For token pruning algorithms, including token pooling(TOME) or dynamic thinning (DynamicViT), it will not change the peak memory of the model because the usage of first block do not decrease. To deploy DynamicViT on MCU, peak memory is over 800kB which exceeds the memory constraint. So we compose low-rank ratio and token number to hypernetwork and evolute in the search space of hypernetworks.
>
> Q3: Co-design approach
>
> A3: We will replace the co-design by co-optimizations in the final version. For algorithm, we first optimize the search space of hypernetworks which is composed of the token number and the low-rank ratio, and then perform architecture search in the updated search space of hypernetworks. For hardware design, we extend the operator library, use overwriting to share buffer and patch embedding decomposition to reduce the memory.
>
> Q4: Supplementary materials
>
> A4: Due to our negligence, we did not submit supplementary materials. We will submit in the final version.
>
>
> Q5: Details of search space evolution
>
> A5:We present the search space evolution framework for effective architecture search of vision transformers within the memory limits. We leverage the one-shot network architecture search method to discover the optimal architecture, where we enlarge the search space by considering low-rank decomposition ratios and token numbers that sizably influences the peak memory during the forward pass.
>
> (1)	total search cost. We deploy the vision transformers with our algorithm-hardware co-optimizations framework in different microcontrollers with different resource constraint. For our method, as 320kB as example, we measured on NVIDIA GeForce RTX 3090 and evolve the search space for 360 GPU hours.
>
> (2)	memory constraints and evaluation. We have the same definition in [2]. Current memory usage contains tembuffer, input tensors and output tensors. Taking MLP block as example, the dimension of this tensor is token number * (embedding dimension * MLP ratio * rank ratio).
>
> (3)	Memory consumption evaluation during search subnet. The equation is peak memory = embedding dimension * ( token number * (1+ MLP ratio) + embedding dimension * MLP ratio * rank ratio)
>
> [1] Chen, Minghao, et al. "Searching the search space of vision transformer."
> [2] MCUNet: Tiny Deep Learning on IoT Devices

---

> > ### Comment · Reviewer_77fa · 2023-08-16
> > **Response to authors**
> >
> > Thank you for your response, which certainly clarified some aspects. However, I still have concerns about the search space evolution algorithm:
> > 1. While search space evolution methods have demonstrated effectiveness, it appears that these algorithms hold more engineering significance in applications with highly constrained memory budgets, albeit potentially lacking in scientific depth. If there's room for a slight relaxation of the memory constraints, it seems that a well-designed manual search space in combination with memory-aware model search might work well.
> > 2. Re-running the search space evolution for each memory constraint requires 360 GPU hours, which is costly.  Could you provide any comments on this? I would highly appreciate a clarification.

---

> > > ### Author Response · Authors · 2023-08-17
> > > **Response to reviewer 77fa**
> > >
> > > We greatly appreciate your thorough review and insightful feedback. Here are our responses to the questions:
> > >
> > > Q1: Scientific depth and manually-designed search space
> > >
> > > A1: Firstly, since the search space significantly influences the trade-off between the memory constraint and performance, selecting the optimal search space is imperative. However, due to difficulty to manually design the model which will explain in the second paragraph, we propose an efficient evolution algorithm to design transformer architectures. Because searching hypernetwork in each search space more than 12000 GPU hours, we design linear fitting function in a small local region including accuracy and the ratio of eligible subnet to estimate the tendency of the search space of hypernetwork.[1] We design the piecewise linear fitting function that only considers top-k nearest neighbors for the search space factor.[2] This enables correct regression during the early stages of training, significantly reducing training time and consequently decreasing the cost of searching for the optimal search space.
> > >
> > > Secondly, since the higher memory usage has better performance,[3,4] our memory constraint should fully utilize MCU memory to ensure optimal performance. But we lack sufficient prior knowledge to select relatively better search space. We are only aware that in search spaces with larger token number and low-rank ratio, the average memory consumption of all networks within the space tends to be higher. Although every search space exists sub-networks whose memory footprint is within the SRAM limit, we cannot acquire accurate estimations of their performance through prior knowledge without conducting actual evaluations. Consequently, this prevents us from manually selecting the optimal search space. Moreover, we introduced a slight relaxation in the memory constraint for 340kB and randomly selected several search spaces since we cannot manually determine which one is the best. Upon comparing these with our automated approach, we observed that the manual method resulted is 64.7% significantly lower performance compared to 73.6%.
> > >
> > > | Token number | Low-rank ratio | Accuracy |
> > > |--------------|----------------|----------|
> > > | 82           | 0.95           | 59.7     |
> > > | 145          | 0.85           | 64.7     |
> > > | 226          | 0.65           | 63.9     |
> > >
> > > I am grateful for your counsel concerning the manually designed search space. We will investigate criteria for assessing the quality of search spaces, aiming to enable manual search space selection while reducing search costs in our future work.
> > >
> > > Q2: training cost
> > >
> > > A2: Firstly, to compare with other transformer NAS methods, AutoFormerV2[2] find better supernet automatically as well and improves the accuracy significantly. However, we follow the experimental setting in AutoFormerV2 and tanks 420 GPU hours which is higher than MCUFormer. Moreover, to compare with other models based on CNN solution ], MCUNet[4] takes 300 GPU hours but has lower accuracy(60.3%). Our method's search costs are comparable to many other transformer NAS techniques and methods for deploying deep models on MCU, making it reasonably affordable. However, our approach achieves significantly superior results in deploying transformers on microcontrollers which accuracy archives 71.1%.
> > >
> > > Secondly, we can modify the search costs by adjusting candidate count, evolution step number and evolution epoch number within the search space. The approximate formula for estimating the costs is as follows: Cost = candidate count * evolution step number * evolution epoch number + searching subnet time. We adjust the factors to compare different search cost and accuracy in the following table. When we change the candidate count and evolution step count, the accuracy can archive 67.1% as well. Users can choose these hyperparameters based on their specific requirements, whether they prioritize higher accuracy or lower search costs.
> > >
> > > | Candidate Count | Evolution Step Count | Accuracy | GPU hours |
> > > |-----------------|----------------------|----------|----------|
> > > | 5               | 5                    | 71.1%    | 360  |
> > > | 3               | 5                    | 68.2%    | 280  |
> > > | 5               | 3                    | 69.5%    | 280  |
> > > | 3               | 3                    | 67.1%    | 250  |
> > >
> > > Thirdly, different MCUs may have the same memory constraint. For STM32F765 and STM32H743, they have the same SRAM(512kB). Therefore, the search costs for finding a transformer network structure that maximizes accuracy for each constraints are acceptable. Moreover, due to MCU manufacturers' design principles, memory constraints are typically limited to a few fixed levels.
> > >
> > > [1] Once for all: Train one network and specialize it for efficient deployment.
> > >
> > > [2] Searching the search space of vision transformer
> > >
> > > [3] MCUNet: Tiny Deep Learning on IoT Devices
> > >
> > > [4] Entropy-Driven Mixed-Precision Quantization for Deep Network Design

---

> > > > ### Comment · Reviewer_77fa · 2023-08-21
> > > > **Response to authors**
> > > >
> > > > Thanks for your further clarification, which addresses some of my concerns. A missing related work[1] can be included in your revision.
> > > >
> > > > I will raise my rating score to borderline accept.
> > > >
> > > >
> > > > [1] SpaceEvo: Hardware-Friendly Search Space Design for Efficient INT8 Inference,  https://arxiv.org/abs/2303.08308

---

> > > > > ### Author Response · Authors · 2023-08-21
> > > > > **Response to Reviewer 77fa**
> > > > >
> > > > > We are grateful for your positive assessment of our paper and your suggestion. Your valuable feedback is sincerely appreciated, and we will refine our paper based on your insights. We will also cite [1] in our revised version.
> > > > >
> > > > > [1] SpaceEvo: Hardware-Friendly Search Space Design for Efficient INT8 Inference

---

### Official Review · Reviewer_67yL · 2023-07-03

**Soundness:** 3 good
**Presentation:** 3 good
**Contribution:** 3 good
**Rating:** 6
**Confidence:** 3

**Summary:**

The paper introduces a hardware-algorithm co-design framework, MCUFormer, that enables the deployment of vision transformers on microcontrollers (MCUs). This framework jointly designs transformer architectures and constructs an inference compiler in order to accommodate the constraints of memory resources. MCUFormer broadens the search space by considering the low-rank decomposition ratio and token numbers, thereby identifying a model topology that offers the highest accuracy within a given resource budget. It also proposes memory scheduling techniques for inference that achieves significant memory reduction, enabling vision transformer on MCU. Experimental results demonstrate MCUFormer achieves higher accuracy on ImageNet compared to previous CNN-based networks on MCU.

**Strengths:**

- Timely contribution: The paper offers a timely contribution to the field by introducing a hardware-algorithm co-design framework capable of deploying vision transformers on MCUs.
- Methodological soundness: The proposed techniques are reasonable and intuitive. They effectively address the challenges of enabling vision transformers on MCUs, given the limited memory. This complements existing research on efficient ViT, which primarily focuses on reducing computational complexity and storage costs.
- Practical Implementation: The authors have operationalized their proposed framework by implementing an inference library. They have measured memory usage and latency on real devices, thereby attesting to the practicality of the framework.
- Proven effectiveness: The evaluation indicates that the proposed framework significantly outperforms CNN-based state-of-the-art networks on MCUs. It delivers a substantial improvement in accuracy (+5.4%) with a similar memory footprint.

**Weaknesses:**

- Lack of ablation study on model size reduction: The evaluation shows the reduction in model size, i.e., flash memory usage, but it doesn't clearly quantify the improvements achieved through the proposed techniques, such as low-rank decomposition. An ablation study could further enhance the paper's quality.
- Missing latency measurements: While the experimental results include memory usage and accuracy metrics achieved by the framework, they do not report on inference latency. As many vision tasks on IoT devices are time-sensitive, results of latency measurement are expected.
- Lack of clarity on search cost: It remains unclear how much search cost is necessary for MCUFormer to support a different combination of models and resource constraints.

**Questions:**

1. Could you please elaborate on the improvements achieved by the proposed techniques, such as low-rank decomposition, in reducing the model size?
2. Why were the results of inference latency not included in the experimental results, and could you provide those measurements?
3. How much search cost is necessary for MCUFormer to accommodate different combinations of models and resource constraints? Could you explain how the search cost was measured or calculated in your study?
4. Is there a plan to make the MCUFormer framework open-source for further research and development in the community?

**Limitations:**

Yes.

---

> ### Author Rebuttal · Authors · 2023-08-10
>
>
> Thank you for careful reading and valuable comments. We will check the paper carefully, and modify the presentation of the ambiguous parts in the final version. We provide answers to the questions as follows:
>
> Q1: How to reduce the memory
>
> A1: Since the memory footprint of vision transformers exceed MCUs constraint and lack of operator library for transformer affects inference on MCUs, existing network compression methods cannot deploy the model on the MCU. And existing NAS model only focus on parameter instead of memory constraint. So we jointly design transformer architectures within the memory resource constraint to reduce the peak memory.
>
> (1)	search algorithms.
>
> Firstly, Low-rank decomposition and token number reduce the dimension of tensor and peak memory. For example, the dimension of input tensor in the MLP block is token number * (embedding dimension * MLP ratio * rank ratio). We first optimize the search space of hypernetworks composed of low-rank ratio and token number to reduce the dimension of tensor and peak memory. But new hypernetwork compositing two factors contains more than 8×10¬¬30 candidate architectures.
>
> Secondly, because of the convergence difficulty of training the hypernetwork we learn the hypernetwork in the fixed search space consisting of given low-rank decomposition ratios and token numbers, and evolve the search space of hypernetwork optimization by predicting the correlation between the task performance and the search space.
>
> Thirdly, due to the strong correlation between the task errors and the continuous choices, we evolve the search space by considering the relation among the performance, memory footprint and the search space factors including token numbers and the low-rank decomposition ratio.
>
> We compared the details of AutoFormer, AutoFormer composed with two factors and MCUFormer on STM32F4 which can refer in the following table.
>
> | Model      | Accuracy | Peak Memory | GPU Hours |
> |------------|----------|-------------|-----------|
> | AutoFormer | 73.7     | 1.24MB      | 200       |
> | Composed   | 68.6     | 218kB       | 600       |
> | MCUFormer  | 71.1     | 218kB       | 360       |
>
> (2)	hardware design. We extend operator library including Layernorm and GeLU since existing library is not feasible for vision transformer quantized inference. In addition, we decrease the receptive field size of parch embedding layer to reduce the memory usage. We also use overwriting sharing input and output tensors buffer to reduce the memory during fully-connect layer inferencing. We compress the model from 3MB to 319kB, fitting the memory constraint. We compare the details of hardware design effect on STM32F7 which can refer in the following tablei.
>
> |                   | Peak Memory | Quantization Approximation | Parch Embedding Decomposition | Overwriting |
> |-------------------|-------------|----------------------------|------------------------------|-------------|
> | Add               | 319kB       | 319kB                      | 319kB                        |             |
> | Unuse             | Cannot Inference | 474kB                  | 484kB                        |             |
>
> Q2: Search cost and results of inference latency
>
> A2: We deploy the vision transformers with our algorithm-hardware co- optimizations framework in different microcontrollers with different resource constraint. We evaluate framework to acquire the accuracy, latency and memory. Our method achieves higher top-1 accuracy and fit the memory constraint.
>
> | MCU      | STM32F4 | STM32F7 | STM32H7 |
> |------------|----------|---------|---------|
> | Constrain  | 256kB    | 320kB   | 512kB   |
> | Accuracy   | 71.1%    | 73.6%   | 74.0%   |
> | Latency    | 706ms    | 641ms   | 798ms   |
> | Memory     | 218kB   | 319kB   | 505kB   |
>
> For our method, 320kB constraint as example, we measured on NVIDIA GeForce RTX 3090 and evolve the search space for 360 GPU hours. We also compare with Autoformer and Autoformer composited with low-rank ratio and token number without evolution which can refer to Q1 Table.
>
> Q3: How the search cost was measured
>
> A3: We measure the training time and search time. Train time include evolution and training the selected hypernetwork.
>
> Q4: MCUFormer framework open-source
>
> A4: Thank you for your interest in our work. If our paper is accepted, our code will be made open source.

---

> > ### Comment · Reviewer_67yL · 2023-08-14
> >
> > Thanks a lot for the authors' response, and I am confident in my rating.

---

> > > ### Author Response · Authors · 2023-08-17
> > > **Response to 67yL**
> > >
> > > We are incredibly grateful for your positive assessment of our paper. Your feedback is highly valued, and we are committed to implementing your suggestions to further elevate the quality of our research.

---

### Official Review · Reviewer_N3hf · 2023-07-06

**Soundness:** 3 good
**Presentation:** 3 good
**Contribution:** 3 good
**Rating:** 6
**Confidence:** 4

**Summary:**

This work proposed a hardware-algorithm co-design method called MCUFormer to enable vision transformer deployment on MCUs with limited memory, which includes both model design and compilation optimization to meet the memory constrain of MCUs. Specifically, the authors utilize one-shot NAS to search the model that fits the memory budget and optimizes the performance at the same time. In addition, they optimize the inference with a set of techniques including operator integerization, patch embedding decomposition, and token overwritting for higher performance. Finally, the experiments shows competitive accuracy improvement given the memory constrain compared to prior CNN based solutions.

**Strengths:**

1) The proposed framework generally addresses the deployment problem of transformers on MCUs for the first time and covers the major toolchain on MCUs including model architecture design, compilation, and operation library implementation.  The experiments confirm the feasibility of the proposed framework and shows promising accuracy improvement compared to prior CNN based counterparts.

2) The proposed NAS framework can be aware of the memory limits of MCUs and constructs a compiling library with efficient memory scheduling to enable model deployment on MCUs.

**Weaknesses:**

1) Some of the key concepts utilized in this paper are confusing. For instance, this paper mentions compiling library and memory scheduling in the proposed framework. Compilation and transformer operation library are totally different things. They need to be clarified in this work especially when MCUs typically do not have a clear AI toolchain definition. As memory scheduling, it is also well defined in computer science and it is a bit misleading when they are utilized as a different consept.

2) There is a lack of ablation study and some of the major technical contributions including the proposed NAS and the model implementation optimization need more evidence.

3) End-to-end performance metrics like fps are not presented.

**Questions:**

1) Since MCUformer includes both cutomized NAS framework and model inference and particularly prior NAS framework i.e. AutoFormer can also produce models that are close to this work, an end-to-end comparison of the two NAS frameworks will be helpful. In addition, given the model optimized with AutoFormer, will it be possible to deploy it on MCUs using the proposed inference optimization techniques.

2) There is always a large design space where performance and accuracy can be compromised. While the resulting transformer models show much higher model accuracy, could you further provide the performance comparison over CNN based solutions.


**Limitations:**

Yes

---

> ### Author Rebuttal · Authors · 2023-08-10
>
>
> Thank you for careful reading and valuable comments. We will check the paper carefully, and modify the presentation of the ambiguous parts in the final version. We provide answers to the questions as follows:
>
>
> Q1: The confusion of compiling library and memory scheduling and the definition of memory scheduling
>
> A1:
>
> For compiling library:
>
> We are sorry for the ambiguity. We extend operator library including Layernorm and GeLU since existing library is not feasible for vision transformer inference. We formulate surrogate equations with fixed-point iterative methods to calculate the output of the square root operator when inferencing Layernorm. We approximate the activation function with quantization by multiplication and softmax function which can both implemented with int8 arithmetics. (We also replace the batchmatmul operator by concatenation, matmul and unpack operators which can implemented on MCU). We will replace the compiling library by operator library in the final version.
>
> For memory scheduling:
>
> We have the same definition in [1] which current memory usage contains tembuffer, input tensors and output tensors. Tembuffer storages the tensors which computed in the previous layers and will use in the following layers. For patch embedding layer, the receptive field of the decomposed patch is contained in the memory as well. We have the different definition with the traditional memory scheduling which focus on task priorities and scheduling order including priority setting, interrupt management, interrupt management and task queuing.
>
> [1] MCUNet: Tiny Deep Learning on IoT Devices
>
> Q2: Ablation study of NAS and the model implementation optimization.
>
> A2: Firstly, we compare our search space evolution method with other selection criteria including randomly sampled search space, the maximal search space and composing the search space selection into the supernet learning which you can refer to Table1 in the article. Secondly, we investigate the influences of the neighborhood size on tendency fitting, and depicts the results for data point sampling with different numbers of nearest neighbors, which you can refer to Table2 in the article. Moreover, to compare, we deploy the initial vision transformer, MCUFormer, the network architecture search replace by AutoFormer and missing one of hardware design techniques on STM32F4(memory constraint is 256kB). You can refer to the following table.
> | Model           | Accuracy | Memory Usage |
> |-----------------|----------|--------------|
> | AutoFormer      | 74.7%    | 3MB          |
> | Without NAS     | 74.7%    | 319kB        |
> | Without overwriting | 71.1% | 373kB        |
> | Without token decomposition | 71.1% | 545kB  |
> | MCUFormer       | 71.1%    | 218kB        |
>
> Q3: End-to-end performance
>
> A3: We deploy the vision transformers with our algorithm-hardware framework in different microcontrollers with different resource constraint. We evaluate framework to acquire the accuracy, latency and memory. Our method achieves higher top-1 accuracy and fits the memory constraint.
>
> |           | STM32F4 | STM32F7 | STM32H7 |
> |-----------|---------|---------|---------|
> | Constrain | 256kB   | 320kB   | 512kB   |
> | Accuracy  | 71.1%   | 73.6%   | 74.0%   |
> | Latency   | 706ms   | 641ms   | 798ms   |
> | Memory    | 218kB   | 319kB   | 505kB   |
>
> Q4: Deploying AutoFormer on MCU
>
> Because the memory footprint of vision transformers exceeds the budget of MCUs and lacks operator library for transformer inference on MCUs, existing network compression methods such as AutoFormer cannot deploy the model on the MCU. Conventional operator library with quantized operators is not feasible for vision transformer inference because of the specific operators including the GeLU activation and layer normalization. So transformer model cannot be inferenced on MCUs. Using our operator library, for STM32F746 320kB constrains, peak memory of AutoFormer reaches 814kB.
>
> Q5: CNN based solutions
>
> Both MCUNet and EMQ are CNN based solution. We evaluate three models on STM32F7 to measure the accuracy and memory usage.
>
> | Model      | MCUNetV1 | MCUNetV2 | EMQ    | MCUFormer |
> |------------|---------|---------|--------|-----------|
> | Accuracy   | 61.8%   | 63.5%   | 68.2%  | 73.6%     |
> | Memory     | 293kB   | 282kB   | 308kB  | 319kB     |

---

> > ### Comment · Reviewer_N3hf · 2023-08-10
> > **The response addressed my questions.**
> >
> > Thanks for the response and I am satisifed with it.

---

> > > ### Author Response · Authors · 2023-08-17
> > > **Response to reviewer N3hf**
> > >
> > > Thank you so much for your valuable feedback on our paper. We greatly appreciate your insights and will incorporate your suggestions to enhance the quality of our work.

---

### Official Review · Reviewer_km2A · 2023-07-06

**Soundness:** 3 good
**Presentation:** 3 good
**Contribution:** 2 fair
**Rating:** 5
**Confidence:** 3

**Summary:**

This paper proposes a co-design method (MCUFormer) that enables the deployment of vision transformers on microcontrollers with limited memory. The method enlarges the search space of vision transformers by considering low-rank decomposition dimensions and patch resolution for memory reduction. The inference library of vision transformers is constructed by scheduling the memory buffer during inference through operator integerization, patch embedding decomposition, and token overwriting.

**Strengths:**

1. The compiling library for Vision Transformers marks a great contribution to the field. This may provide valuable insights and tools for future research within this domain.
2. The results showcased in Table 3 are remarkable.

**Weaknesses:**

While the Compiling Library Construction does provide an element of novelty, the search algorithm essentially leverages the AutoFormer. A deeper discussion and analysis would be beneficial, particularly low-rank decomposition dimensions and patch resolution for memory reduction.



**Questions:**

See weakness.

**Limitations:**

The authors have addressed the limitations.

---

> ### Author Rebuttal · Authors · 2023-08-10
>
>
> Thank you for careful reading and valuable comments. We will check the paper carefully, and modify the presentation of the ambiguous parts in the final version. We provide answers to the questions as follows:
>
> Q: The role of low-rank decomposition dimensions and patch resolution for memory reduction. \
> A: Because the memory footprint of vision transformers exceeds the budget of MCUs and lacks operator library for transformer inference on MCUs, existing network compression methods such as AutoFormer cannot deploy the model on the MCU. Without quantization and optimization, for STM32F746 320kB constraint, peak memory of AutoFormer reaches over 3MB which is 9.7 times to the constraint. Our technical contribution is listed as follows. \
> (1)	search algorithm. We first optimize the search space of hypernetworks that is composed of low-rank ratio and token number to reduce the dimension of tensor and peak memory. Due to the extremely high training cost, we evolve the search space by considering the relation among the performance, memory footprint and the search space factors including token numbers and the low-rank decomposition ratio. According to the score function, we get the final candidate hypernetwork and search in this hypernetwork. \
> (2)	hardware design. We extend operator library including Layernorm and GeLU since existing library is not feasible for vision transformer inference. We decompose the patch embedding layer by transforming the convolution with large receptive field to multiple convolution operations with small receptive field to reduce the memory usage. To add the results from multiple convolution operations, we leverage small memory buffer to record the intermediate value of accumulation. \
> For STM32F746 320kB constrains, the peak memory of MCUFormer is 319kB, accuracy is 73.6% and latency is 641ms. For deploying AutoFormer on MCU, using extended operator library, peak memory is 1.24MB and the accuracy is 74.7%.

---

### Decision · Program_Chairs · 2023-09-21

**Decision:**

Accept (poster)

**Comment:**

The paper builds a Vision Transformer (MCUFormer) that is able to run under very limited memory constraints on microcontrollers, and retain good accuracy. The do this, the paper proposes a generalized one-shot NAS method, alongside additional techniques to save memory. The paper received unanimous accept recommendations, however, 3/5 of these were only "Borderline". The reviewers highlight the importance of the work, as well as the strong results. The additional analysis in the rebuttal was also appreciated. One valid concern could be that the MCUFormer combines a combination of ML and engineering tricks, and it is not clear what the key generalizable scientific takeaway of the paper is. However, achieving strong results on a practical problem can still be of great value to the community, so I recommend acceptance.